# Performance of COVID-19 associated symptoms and temperature checking as a screening tool for SARS-CoV-2 infection

Benjamin Demah Nuertey[1,2,3]*, Kwame Ekremet[1], Abdul-Rashid Haidallah[1], Kareem Mumuni[1,4], Joyce Addai[5], Rosemary Ivy E. Attibu[1,3], Michael C. Damah[1,6], Elvis Duorinaa[1,7], Anwar Sadat Seidu[1,3], Victor C. Adongo[1,8], Richard Kujo Adatsi[1,8], Hisyovi Caedenas Suri[1,9], Abass Abdul-Karim Komei[10], Braimah Baba Abubakari[11,12], Enoch Weyori[10], Emmanuel Allegye-Cudjoe[13], Augustina Sylverken[14,15], Michael Owusu[15,16], Richard O. Phillips[15]

1 Tamale Teaching Hospital, COVID-19 Management Team, Accra, Ghana, 2 Community Health Department, University of Ghana Medical School, Accra, Ghana, 3 Public Health Department, Tamale Teaching Hospital, Tamale, Ghana, 4 Department of Obstetric and Gynaecology, University of Ghana Medical School, Accra, Ghana, 5 Department of Medicine, Korle-Bu teaching Hospital, Accra, Ghana, 6 Pharmacy Department, Tamale Teaching Hospital, Tamale, Ghana, 7 Department of Surgery, Tamale Teaching Hospital, Tamale, Ghana, 8 Laboratory Department, Tamale Teaching Hospital, Tamale, Ghana, 9 Intensive Care Unit, Tamale Teaching Hospital, Tamale, Ghana, 10 Zonal Public Health Reference Laboratory, Tamale, Ghana, 11 Regional Health Directorate, Northern Region, Tamale, Ghana, 12 School of Medical Sciences, University for development studies, Tamale, Ghana, 13 Veterinary Service Laboratory, Pong-Tamale, Ghana, 14 Department of Theoretical and Applied Biology, Kwame Nkrumah University of Science and Technology, Kumasi, Ghana, 15 Kumasi Centre for Collaborative Research, Kwame Nkrumah University of Science and Technology, Kumasi, Ghana, 16 Department of Medical Diagnostics, Kwame Nkrumah University of Science and Technology, Kumasi, Ghana

* ben.nuertey@gmail.com

## Abstract

## Introduction

Coronavirus disease-19 (COVID-19), which started in late December, 2019, has spread to affect 216 countries and territories around the world. Globally, the number of cases of SARS-CoV-2 infection has been growing exponentially. There is pressure on countries to flatten the curves and break transmission. Most countries are practicing partial or total lock-down, vaccination, massive education on hygiene, social distancing, isolation of cases, quarantine of exposed and various screening approaches such as temperature and symptom-based screening to break the transmission. Some studies outside Africa have found the screening for fever using non-contact thermometers to lack good sensitivity for detecting SARS-CoV-2 infection. The aim of this study was to determine the usefulness of clinical symptoms in accurately predicting a final diagnosis of COVID-19 disease in the Ghanaian setting.

## Method

The study analysed screening and test data of COVID-19 suspected, probable and contacts for the months of March to August 2020. A total of 1,986 participants presenting to Tamale

**Data Availability Statement:** All data files are available from the Tamale Teaching hospital,

Research department database. Also publicly available via OSF: https://osf.io/fz82k/.

**Funding:** The authors received no specific funding for this work.

**Competing interests:** The authors have declared that no competing interest exist.

Teaching hospital were included in the study. Logistic regression and receiver operator characteristics (ROC) analysis were carried out.

## Results

Overall SARS-CoV-2 positivity rate was 16.8%. Those with symptoms had significantly higher positivity rate (21.6%) compared with asymptomatic (17.0%) [chi-squared 15.5, p-value, <0.001]. Patients that were positive for SARS-CoV-2 were 5.9 [3.9–8.8] times more likely to have loss of sense of smell and 5.9 [3.8–9.3] times more likely to having loss of sense of taste. Using history of fever as a screening tool correctly picked up only 14.8% of all true positives of SARS-CoV-2 infection and failed to pick up 86.2% of positive cases. Using cough alone would detect 22.4% and miss 87.6%. Non-contact thermometer used alone, as a screening tool for COVID-19 at a cut-off of 37.8 would only pick 4.8% of positive SARS-CoV-2 infected patients.

## Conclusion

The use of fever alone or other symptoms individually [or in combination] as a screening tool for SARS-CoV-2 infection is not worthwhile based on ROC analysis. Use of temperature check as a COVID-19 screening tool to allow people into public space irrespective of the temperature cut-off is of little benefit in diagnosing infected persons. We recommend the use of facemask, hand hygiene, social distancing as effective means of preventing infection.

## Introduction

Coronavirus disease-19 (COVID-19), which started in late December, 2019, has spread to affect 216 countries and territories around the world [1, 2]. More than 151 million of the world population are affected with over 3.1 million recorded deaths as at end of April, 2021 [1, 2]. All African countries have reported cases of COVID-19. However in Africa, the outbreak had been relatively slow in reaching all countries [1, 2]. As at end of April 2021, Africa recorded over 4.5. million cases and 121,000 deaths [1]. Also, the death rates and number of cases per million population in Africa has been relatively low compared to other parts of the world. As the number of cases of SARS-CoV-2 infection is growing exponentially, there is pressure on most countries to flatten their curves and break transmission. Most countries are practicing partial or total lockdown, vaccinations, massive education on hygiene, social distancing, isolation of cases, quarantine of exposed and various screening approaches based on the clinical spectrum of COVID-19 to break the transmission of SARS-CoV-2 [3–7].

Clinical spectrum of SARS-CoV-2 infection is known to range from asymptomatic infections to death [8–10]. Although, symptoms associated with COVID-19 are non-specific, initial 710 cases of COVID-19 had fever (98%), cough (76%), Headache (8%), haemoptysis (5%), diarrhoea (3%) [9, 11]. Subsequent clinical studies from most part of the world identified fever, dry cough, fatigue, myalgia, dyspnoea, diarrhoea, loss of smell, loss of taste, nausea and vomiting as common symptoms associated with SARS-CoV-2 infection [12, 13]. This made these set of symptoms to be characterised as COVID-19 associated symptoms and have been part of screening tools for SARS-CoV-2 infection in most parts of Africa. However, these are not the gold standard for SARS-CoV-2 detection.

Real time reverse transcription-polymerase chain reaction (rRT-PCR) on respiratory tract sample remain the reference standard for diagnosing SARS-CoV-2 infection which causes COVID-19 [14, 15]. Diagnostic tests such as computed tomography (CT) scan, chest x-rays, temperature and symptoms based check have been described as screening tools for COVID-19 [16–21]. Several pre-analytic and analytic errors can affect diagnostic accuracy of PCR and screening tests [22, 23]. The clinical, social, psychological and economic consequence of errors in diagnostic tests are considerably high. This consequences are further amplified in the cases of highly infectious disease outbreaks such as COVID-19 [22]. The repercussions of a false positive or a false negative results goes beyond the individual. It has the potential of endangering the overall public health response to the outbreak. PCR test in most parts of the world have long turnaround time preventing its extensive use in case scenarios where a rapid tests results is required. From the time of taking of sample, transportation, test and receipt of results could range from about 6 hours to several days, but typically ranges between two to six days depending on the workload on the testing facility [24, 25].

In the absence of specific therapeutic medications, it is essential to rapidly diagnose and isolate suspected SARS-CoV-2 infected persons [26] which is our best chance at eradicating the SARS-CoV-2. Most health facilities, shops, restaurants, airports and public places are relying on temperature screening among other symptoms and infection prevention strategies such as facemask use, hand hygiene and social distancing to prevent SARS-CoV-2 infection in public places. Temperature screening approaches in most part of Africa uses thermal scanners, infrared/ non-contact thermometers. However, most cases of SARS-CoV-2 infection are increasingly becoming asymptomatic or afebrile [27–29]. This questions the accuracy and cost effectiveness of using these symptom-based screening methods.

Also, the sensitivity and specificity of temperature-based methods in Africa are poorly understood. Studies have shown that, screening for COVID-19 using fever/ temperature checking lack the sensitivity to detect SARS-CoV-2 infection and may have negligible value for controlling the COVID-19 pandemic [30, 31]. Though the prevalence of alteration of taste and or smell ranges between 41–46% of study samples as reported by systematic reviews and meta-analysis [32], its diagnostic value in screening for SARS-CoV-2 is of limited value. The aim of this study was to determine the test performance of temperature-based screening and other symptom-based screening methods of COVID-19 in an African setting.

## Materials and methods

### Patient and data source

This was a retrospective study that analysed clinical and laboratory test results of all persons who had a COVID-19 test between March 2020 and August 2020. The primary data was aggregated data on patient and samples sent by the Tamale Teaching Hospital COVID-19 case management team to three testing facilities.

### Setting

Until the end of April 2020, the Kumasi Centre for Collaborative Research in Tropical Medicine (KCCR) was the only testing site providing SARS-CoV-2 rRT-PCR for the middle and northern sector of Ghana. The Zonal public health reference laboratory, Tamale (ZPHRLT) and the Central Veterinary service laboratory (CVSL) were later equipped to also carry out testing by 1st of May, 2020. Samples taken from all health facilities within the middle and northern sector are transported to KCCR or the ZPHRLT under standard transport conditions. Samples were accompanied with a case investigation form (S1 Text).

## Sample taking, transport and testing

The clinicians, laboratory staff and disease control officers collected pre-testing demographic and clinical data. In most instances, nasopharyngeal and or oropharyngeal swabs was collected and placed in cryotubes containing viral transport media (VTM) and stored immediately on ice or refrigerated (4–8 °C) and transported within 24 hours to the testing sites. There were multiple instances where sputum samples were collected into sputum containers due to lack of swabs and or VTM. For the months of June through to August, sputum sample became the standard practice due to VTM shortage and nasopharyngeal swabs were reserved for children or patients who could not provide sputum samples. The average distance travelled from where samples were taken to the testing centre, (KCCR) in Kumasi is about 393Km. The ZPHRLT is however located within the premises of the Tamale teaching hospital. Fig 1 is a map of Ghana that illustrates the distances of the testing centres from the Tamale teaching hospital.

Typically, samples are triple packaged and kept in sample carriers with cold packs. Samples are transported to the testing centre on a dedicated ambulance or hospital vehicles to KCCR. Real time reverse transcriptase-polymerase chain reaction (rRT-PCR) were carried out targeting ribonucleic acid (RNA) dependent RNA polymerase to detect the presence of SARS-CoV-2 and pan coronavirus (pan-CoV) [33]. RT-PCR was carried out following standard procedures [34]. A test was declared positive if PCR detected both SARS-CoV-2 and PanCoV. The test was repeated if only SARS-CoV-2 was detected while pan-CoV was not detected. It was declared negative for SARS-CoV-2 if SARS-CoV-2 was not detected by RT-PCR. Samples with SARS-CoV-2 RT-PCR cycle threshold (CT) value under 40 were considered positive.

## Variables

The primary outcome was testing positive for SARS-CoV-2. Independent demographic variables were sex (male or female) and age (grouped 0–9, 10–19, 20–29, 30–39, 40–49, 50–59 and 60 and above). We analysed data on clinical status at time of sample taking such as symptomatic or asymptomatic, temperature measured using non-contact thermometers, history of travel outside Ghana within the last fourteen days to the date of onset of symptoms or date sample was taken. The following symptoms were analysed as categorical variables; history of fever, general weakness, cough, sore throat, runny nose, shortness of breath, diarrhoea, headache, pain (muscular, chest, abdominal and joint pains), anosmia (loss of sensation of smell) and ageusia (loss of sensation of taste).

## Ethical consideration

Ethical clearance for this study was obtained from the Tamale Teaching Hospital Ethical review committee reference ID- TTHERC/30/09/20/07. This retrospective study made use of testing records and data collected as part of the COVID-19 response team in the Tamale Teaching hospital. Anonymized retrospective data was used and deposited in an open access repository [35].

## Statistical analysis

Microsoft Excel data was cleaned and exported to STATA version 14 for analysis. Follow-up tests for already confirmed positives under treatment were excluded from the analysis so as to avoid double counting in the analysis. Cross tabulations were used to determine frequency and proportions of positive and negative SARS-CoV-2 among the various variables. Logistic regression was conducted to determine factors that are independently associated with SARS-CoV-2 infection to include in the receiver operator characteristic analysis. First, each variable

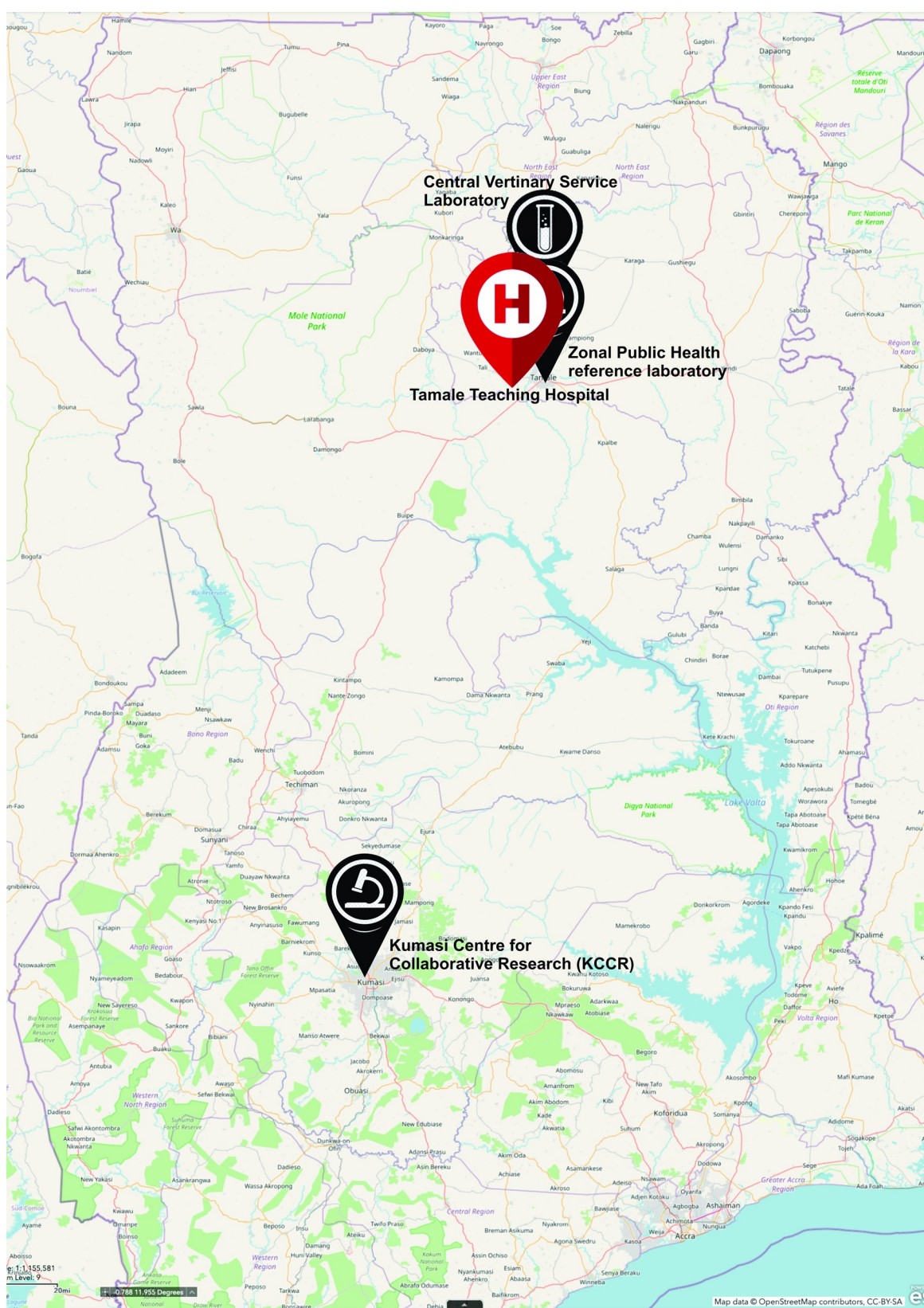

**Fig 1. Map of Ghana showing the Tamale teaching hospital and the SARS-CoV-2 PCR testing facilities used in this study.**

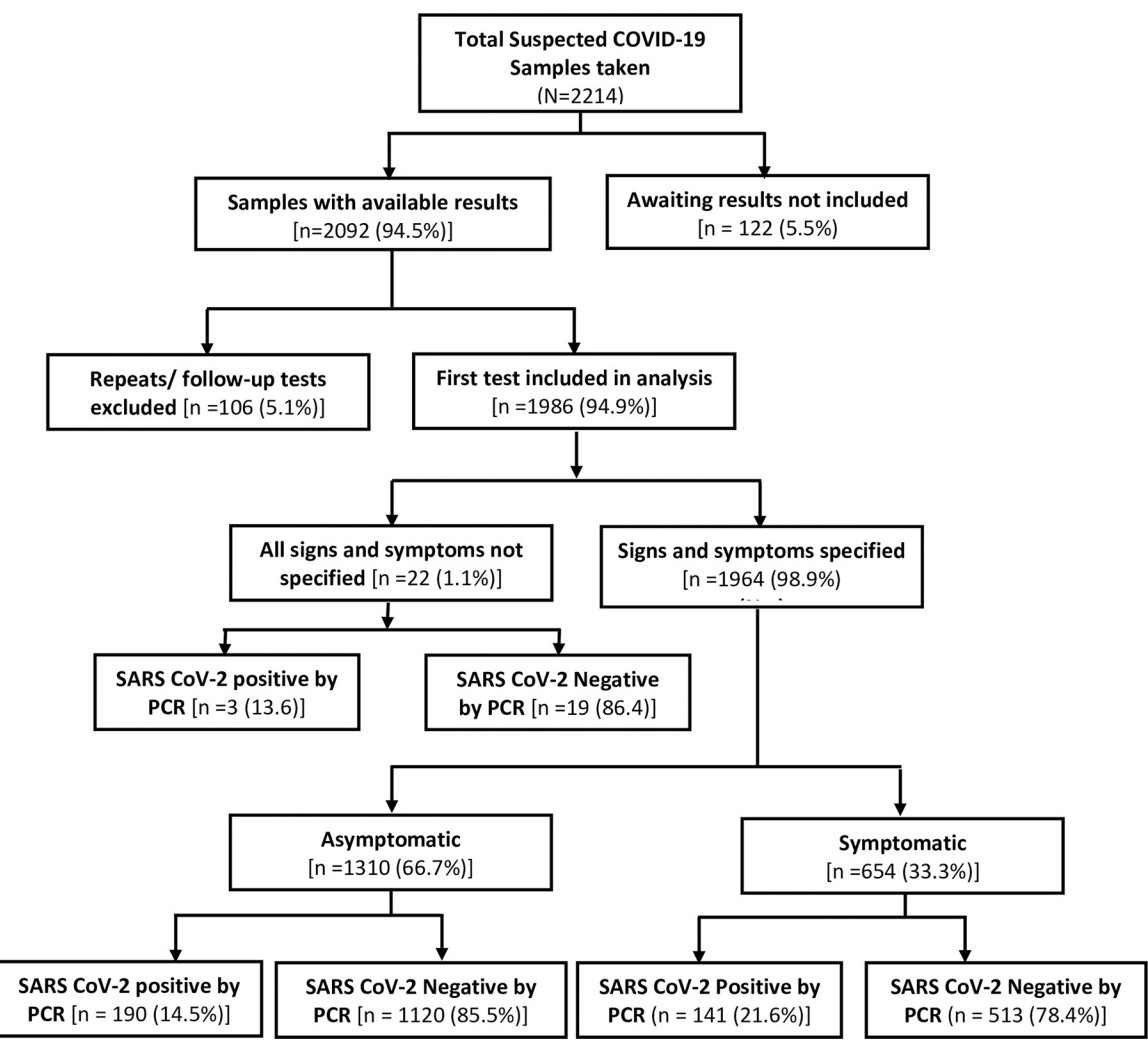

**Fig 2. Flow chart showing data selected for analysis.**

was considered a candidate for inclusion in a multiple logistic regression if the p-value was less than 0.05 in the univariate logistic regression. Fever and sore throat were considered a-priori as independent factor and were eligible for inclusion irrespective of their p-value. The multiple logistic regression was carried out adjusting for age, sex and history of travel outside the country within the last fourteen days. Variables whose p-value were less than 0.05 in the multiple logistic regression were included in the receiver operator characteristic (ROC) analysis. The ROC analysis reported sensitivity, specificity, area under curve, likelihood ratio positive and likelihood ratio negative. ROC curve of sensitivity versus 1-specificity were plotted for all factors. Traditionally, the area under a curve of a worthless test is 0.5 [36]. This means the test includes the point of 50% sensitivity and 50% specificity. Such a test has a diagnostic ability equivalent to flipping a coin [36]. This was used to make a decision on the usefulness of the symptoms-based screening as a tool for picking up SARS-CoV-2 infection. A screening test was deemed worthless if the area under the curve is around 0.5. The performance of a screening test is deemed to be desirable if Area Under Curve (AUC) is $0.7 \geq \text{AUC} \leq 0.8$, excellent if $0.8 < \text{AUC} \leq 0.9$ and outstanding if AUC is $> 90$ [37]. Fig 2 is a flow chart showing data selected for analysis.

## Results

### Background characteristics

Table 1 displays the characteristic of the various variables with respect to the proportion who tested positive or negative for SARS-CoV-2. In all, 1,986 suspected COVID19 patient tests were included in the analysis. This number excludes repeats or follow-up tests. The overall positivity rate was 16.8% for the period March 2020 through to August 2020. Patients with symptoms have higher positivity rate (21.6%) compared with those that were asymptomatic (17.0%) [chi square ($\chi^2$) 15.5, p-value, <0.001]. With regards to sex, 58.4% of the positives were males. The 20–29 years old group had the highest positivity rate of [17.7%, n = 115] followed by the 30–39 year group [16.1%, n = 130].

### SARS-CoV-2 and its associated factors

Table 2 displays the results of logistic regression of factors associated with testing positive for SARS-CoV-2. Loss of sense of smell (anosmia) and loss of sense of taste (ageusia) were independently more likely to be associated with testing positive for SARS-CoV-2. Patients that were positive for SARS-CoV-2 were 5.9 [3.9–8.8] times more likely to have loss of sense of smell and 5.9 [3.8–9.3] times more likely to having loss of sense of taste.

Adjusting for age and sex in a multiple logistic regression analysis, fever, sore throat, general weakness, cough, loss of smell and loss of taste were found to have significant association with the likelihood of testing positive for SARS-CoV-2.

### Receiver operator characteristics of factors associated with COVID19 disease

Factors found to be associated with SARCoV2 such as fever, sore throat, general weakness, cough, loss of smell and loss of taste were included in receiver operator characteristic analysis to determine the sensitivity, specificity and area under the curve if used alone as a screening tool. Table 3 displays the receiver operator characteristics of factors independently associated with SARS-CoV-2 infection. Using history of fever as a screening tool would correctly detect 14.8% whilst cough alone would detect 22.4% of all true positives of SARS-CoV-2 infection.

Checking of temperature when used alone as a screening tool for COVID-19 was associated with the worse test performance. For example, using a temperature cut off of 37.8 would only pick 4.2% of positive SARS-CoV-2 infected patients. Other temperature cut-offs and their respective sensitivity, specificity and likelihood ratios are as shown in Table 4.

Fig 3 shows that, the area under the curve for all classifiers is statistically equivalent and very close to 0.5 which is the area under the curve for a worthless test indicating that using any of the symptom base screening tools alone, such as fever, cough, loss of smell, loss of taste, headache, bodily pain, sore throat and general weakness is not significantly different from a reference worthless test. Even at given age ranges, such as shown in S1 Table, temperature screening fails to give a desirable performance.

Also using a non-contact thermometer for the checking of temperature as a screening tool irrespective of the temperature cut-off used perfectly correlates with a worthless test in deciding if a person is infected with SARS-CoV-2 infection. Table 4 and Fig 4 show the receiver operator characteristics of temperature measurement as a screening tool for COVID-19.

### Combination of symptoms

We used a combination of symptoms such as the case definition for screening for SARS-CoV-2 infection in the Tamale Teaching Hospital. This is an adaptation of the World health

**Table 1. Background characteristics of study participants.**

| Characteristics | | SARS-CoV-2 by PCR | | |
| --- | --- | --- | --- | --- |
| | Total | Negative | Positive | |
| | N | n (%) | n (%) | χ² (p-value) |
| **All** | 1,986 | 1,652 (83.2) | 334 (16.8) | |
| **Symptoms** | | | | |
| Asymptomatic | 1,310 | 1,120 (68.6) | 190 (57.4) | 15.50 (<0.001) |
| Symptomatic | 654 | 513 (31.4) | 141 (41.6) | |
| **Sex** | | | | |
| Male | 1,168 | 973 (59.0) | 195 (58.4) | 0.050 (0.82) |
| Female | 814 | 675 (40.1) | 139 (41.6) | |
| **Age groups (years)** | | | | |
| 0–9 | 70 | 65 (3.9) | 5 (1.5) | 8.13 (0.23) |
| 10–19 | 63 | 54 (3.3) | 9 (2.7) | |
| 20–29 | 650 | 535 (32.4) | 115 (34.4) | |
| 30–39 | 805 | 675 (40.9) | 130 (38.9) | |
| 40–49 | 215 | 178 (10.8) | 37 (11.1) | |
| 50–59 | 98 | 76 (4.6) | 22 (6.6) | |
| 60 and above | 84 | 68 (4.1) | 16 (4.8) | |
| Measured Temperature | | | | |
| 37.2˚C and below | 1,867 | 1,555 (94.1) | 312 (93.4) | 0.25 (0.62) |
| 37.3˚C and above | 119 | 97 (5.9) | 22 (6.6) | |
| **Self-reported Fever** | | | | |
| No fever | 1,779 | 1,497 (91.2) | 282 (85.2) | 11.04 (0.001) |
| Had fever | 194 | 145 (8.8) | 49 (14.8) | |
| **General bodily weakness** | | | | |
| No bodily weakness | 1,765 | 1,487 (90.6) | 278 (84.0) | 12.62 (<0.001) |
| Had bodily weakness | 208 | 155 (9.4) | 53 (16.0) | |
| **Had history of cough** | | | | |
| No cough | 1,635 | 1,378 (83.9) | 257 (77.6) | 7.65 (0.006) |
| Presence of cough | 338 | 264 (16.1) | 74 (22.4) | |
| **Sore throat** | | | | |
| No sore throat | 1,771 | 1,483 (90.9) | 288 (87.0) | 4.65 (0.031) |
| Had sore throat | 192 | 149 (9.1) | 43 (13.0) | |
| **Runny nose** | | | | |
| No runny nose | 1,831 | 1,529 (93.1) | 302 (91.2) | 1.46 (0.227) |
| Had runny nose | 142 | 113 (6.9) | 29 (8.8) | |
| **Shortness of Breath** | | | | |
| No shortness of breath | 1,816 | 1,517 (93.1) | 299 (90.3) | 3.15 (0.076) |
| Had shortness of breath | 144 | 112 (6.9) | 32 (9.7) | |
| **Diarrhoea** | | | | |
| No diarrhoea | 1,933 | 1,608 (97.9) | 325 (98.2) | 0.09 (0.761) |
| Had diarrhoea | 40 | 34 (2.1) | 6 (1.8) | |
| **Headache** | | | | |
| No headache | 1,699 | 1,429 (87.0) | 270 (81.6) | 6.86 (0.009) |
| Had headache | 274 | 213 (13.0) | 61 (18.4) | |
| **Muscular, Chest, Abdominal and joint pains** | | | | |
| No pain | 1,818 | 1,525 (92.3) | 293 (87.7) | 7.55 (0.006) |
| Had pain | 168 | 127 (7.7) | 41 (12.3) | |

(*Continued*)

**Table 1.** (Continued)

| Characteristics | | SARS-CoV-2 by PCR | | |
| --- | --- | --- | --- | --- |
| | Total | Negative | Positive | |
| | N | n (%) | n (%) | χ² (p-value) |
| **Loss of sensation of smell (anosmia)** | | | | |
| Normal sense of smell | 1,702 | 1,435 (96.6) | 267 (82.7) | 93.82 (<0.001) |
| Anosmia | 106 | 50 (3.4) | 56 (17.3) | |
| **Loss of sensation of Taste (ageusia)** | | | | |
| Normal sense of taste | 1,724 | 1,446 (97.4) | 278 (86.1) | 76.54 (<0.001) |
| Loss of sense of taste | 84 | 39 (2.6) | 45 (13.9) | |

organisation case definition for COVID-19. S1 Test displays the combination of symptoms used. Symptoms such as fever, cough, sneezing, sore throat, and runny nose were part of a clinical criteria 1 and having at least two of these symptoms met criteria 1. Also, other symptoms such as difficulty in breathing, altered sense of smell and taste were part of clinical criteria 2 and having at least one of these symptoms met criteria 2. The ROC of such a combination of symptoms were obtained. S2 Table shows the proportion of participants meeting the case definition. Among those who met clinical criteria 1, 195 (77.1%) tested negative while only 58 (2.9%) tested positive. Also, among those meeting criteria 2, 150 (65.5%) tested negative and only 79 (34.5%) tested positive. Table 5 displays the ROC of combination of symptoms. Even with the combination of symptoms, the area under the curve and sensitivity is low. Clinical criteria 1 is associated with a sensitivity of 17.5% and a specificity of 88.1%. Also, clinical criteria 2 was associated with a sensitivity of 24.7% and a specificity of 89.7%. A combination of criteria 1 and 2 gave a slightly better sensitivity of 33.8% and a specificity of 81.9%. S1 Fig displays the receiver operator characteristic curve of clinical combination of symptoms. The combination of symptoms as exemplified by clinical criteria 1 or 2 missed 66.3% of all positive cases.

## Discussion

We have shown that, patients with symptoms have higher positivity rate (21.6%) compared with those that were asymptomatic (17.0%) [chi square ($\chi^2$) 15.5, p-value, <0.001]. The 20–29 years old group had the highest positivity rate of [17.7%, n = 115] followed by the 30–39 year group [16.1%, n = 130]. Patients that were positive for SARS-CoV-2 were 5.9 [3.9–8.8] times more likely to have loss of sense of smell and 5.9 [3.8–9.3] times more likely to having loss of sense of taste. Also, fever, sore throat, general weakness, cough, loss of smell and loss of taste were found to have significant association with the likelihood of testing positive for SARS-CoV-2. Using history of fever as a screening tool would correctly detect 14.8% whilst cough alone would detect 22.4% of all true positives of SARS-CoV-2 infection. Using a non-contact thermometer for the checking of temperature at a cut off of 37.8 would only pick 4.8% of positive SARS-CoV-2 infected patients. This observation is true irrespective of segregation of results by age. Also, findings suggest that, the use of fever alone or other symptoms individually [or in combination] as a screening tool for SARS-CoV-2 infection is not worthwhile based on ROC analysis.

With COVID-19 surge across many countries, most countries are relying on rapid screening techniques such as temperature check and symptoms based triaging methods relying on fever to restrict the movement of individuals into public places. Temperature or fever check takes place at hospitals, airports, bus terminals, restaurants, shops among many other public places. While the checking of such may be beneficial for some other conditions, this practice

**Table 2. Logistic regression of factors independently associate with positive SARS-CoV-2 infection.**

| Independent Factors | Univariate logistic regression | | Multivariate logistic regression | |
|---|---|---|---|---|
| | OR [95% CI] | P-value | *AOR [95% CI] | P-value |
| **Having Symptoms** | | | | |
| No symptom | - | | | |
| Having symptom(s) | 1.6 [1.3–2.1] | <0.001 | 1.6 [1.3–2.1] | <0.001 |
| Measured temperature | | | | |
| Below 37.3 °C | - | | | |
| Below 37.3 °C | 1.1 [0.7–1.8] | 0.616 | - | |
| **Fever** | | | | |
| No Fever | - | | | |
| Had Fever | 1.8 [1.3–2.5] | 0.001 | 1.8 [1.3–2.5] | 0.001 |
| **General weakness** | | | | |
| No general weakness | - | | | |
| Had general weakness | 1.8 [1.3–2.5] | <0.001 | 1.8 [1.3–2.5] | 0.001 |
| **Cough** | | | | |
| No cough | - | | | |
| Coughing | 1.5 [1.1–2.0] | 0.006 | 1.5 [1.1–2.0] | 0.005 |
| **Sore throat** | | | | |
| No sore throat | - | | | |
| Had sore throat | 1.5 [1.0–2.1] | 0.03 | 1.5 [1.0–2.1] | 0.04 |
| **Runny nose** | | | | |
| No runny nose | - | | | |
| Had runny nose | 1.3 [0.8–2.0] | 0.23 | - | |
| **Shortness of Breath** | | | | |
| No shortness of breath | - | | | |
| Had shortness of breath | 1.5 [1.0–2.2] | 0.08 | - | |
| **Diarrhoea** | | | | |
| No diarrhoea | - | | | |
| Had diarrhoea | 0.9 [0.4–2.1] | 0.76 | - | |
| **Headache** | | | | |
| No headache | - | | | |
| Had headache | 1.5 [1.1–2.1] | 0.009 | 1.5 [1.1–2.0] | 0.01 |
| **Nausea and Vomiting** | | | | |
| No nausea and vomiting | - | | | |
| Had nausea and vomiting | 1.4 [0.7–2.7] | 0.32 | - | |
| **All forms of bodily pains** | | | | |
| No bodily pains | - | | | |
| Had bodily pains | 1.7 [1.2–2.4] | 0.006 | 1.6 [1.1–2.3] | 0.01 |
| **Loss of sensation of smell (anosmia)** | | | | |
| Normal sense of smell | - | | | |
| Anosmia | 6.0 [4.0–9.0] | <0.001 | 5.9 [3.9–8.8] | <0.001 |
| **Loss of sensation of Taste (ageusia)** | | | | |
| Normal sense of taste | - | | | |
| Ageusia | 6.0 [3.8–9.4] | <0.001 | 5.9 [3.8–9.3] | <0.001 |

*AOR of multiple logistic regression adjusting for age and sex.

**Table 3. Receiver operator characteristics of factors.**

| Classifier | Sensitivity | Specificity | Correctly classified | LR+ | LR- | ROC Area [95% CI]* |
|---|---|---|---|---|---|---|
| Fever | 14.8% | 91.2% | 78.4% | 1.8 | 0.93 | 0.53 [0.51–0.55] |
| General weakness | 16.0% | 90.6% | 78.1% | 1.7 | 0.93 | 0.53 [0.51–0.55] |
| Cough | 22.4% | 83.9% | 73.6% | 1.4 | 0.93 | 0.53 [0.51–0.56] |
| Sore throat | 13.0% | 90.9% | 77.7% | 1.4 | 0.96 | 0.52 [0.50–0.54] |
| Headache | 18.4% | 87.0% | 75.5% | 1.4 | 0.94 | 0.53 [0.51–0.55] |
| Pain (chest/ joint/ abdominal/ muscular) | 12.3% | 92.3% | 78.9% | 1.6 | 0.95 | 0.52 [0.50–0.54] |
| Loss of smell (anosmia) | 17.3% | 96.6% | 82.5% | 5.1 | 0.86 | 0.57 [0.55–0.59] |
| Loss of taste (ageusia) | 13.9% | 97.4% | 82.5% | 5.3 | 0.88 | 0.56 [0.54–0.58] |

was not a useful screening tool in discriminating people who were suspected to have SARS-CoV-2 infection from getting to public places. Here, we used temperature measured and history of fever within the last fourteen days and show that the practice was worthless test. Temperature check appeared to be worse a screening tool when compared with asking for history of fever. This is because, not all who have fever are picked during screening for temperature [38, 39]. Temperature screen alone without asking of fever underestimates the true proportion of febrile patients and this means that, temperature alone has worse sensitivity for picking SARS-CoV-2 infection compared to asking of history of fever. A study in a tropical region concluded that, infrared handheld thermoscope should not be used for the checking of fever in tropical conditions due to their low sensitivity of 29.4% in picking up those who have fever compared with self-reported fever which has a sensitivity of 88.2% [40]. Mathematical models of screening of travellers have found that, screening would miss half of infected people [41, 42]. Our findings suggest, that using temperature would miss at least 84% of infected people. This observation was true irrespective of age of participants. The use of temperature as screening tool analysed by age suggests that, temperature checking performed poorly in younger age group (less than 20) compared with the 20–39-year-old group. This same findings were observed by others who found that, screening for fever is not sensitive enough to detect the vast majority of SARS-CoV-2 in those age between 18 and 25 [43] and that mass screening with for COVID-19 with non-contact infra-red thermometers does not work [44]. This suggest that, these symptom-based approaches are imperfect barriers to preventing the spread of COVID-19 in this part of the world where most of our cases are increasingly asymptomatic.

**Table 4. Receiver operator characteristics of temperature check.**

| Temperature (ºC) cut-point >/ = | Sensitivity | Specificity | Correctly classified | LR+ | LR- |
|---|---|---|---|---|---|
| 36.7 | 23.7% | 76.5% | 67.6% | 1.00 | 1.00 |
| 37.3 | 6.6% | 94.1% | 79.4% | 1.12 | 0.99 |
| 37.5 | 6.0% | 95.6% | 80.6% | 1.37 | 0.98 |
| 37.7 | 4.8% | 96.2% | 80.8% | 1.26 | 0.99 |
| 37.8 | 4.2% | 96.% | 81.0% | 1.19 | 0.99 |
| 38.0 | 2.7% | 96.9% | 81.0% | 0.86 | 1.00 |
| 38.2 | 1.5% | 97.5% | 81.4% | 0.60 | 1.01 |
| 38.5 | 0.9% | 98.3% | 81.9% | 0.53 | 1.01 |
| 38.7 | 0.9% | 98.8% | 82.2% | 0.67 | 1.00 |
| 39.0 | 0.3% | 99.0% | 82.4% | 0.29 | 1.01 |

ROC area under curve of using temperature check is 0.48, 95% [CI = 0.45–0.52].

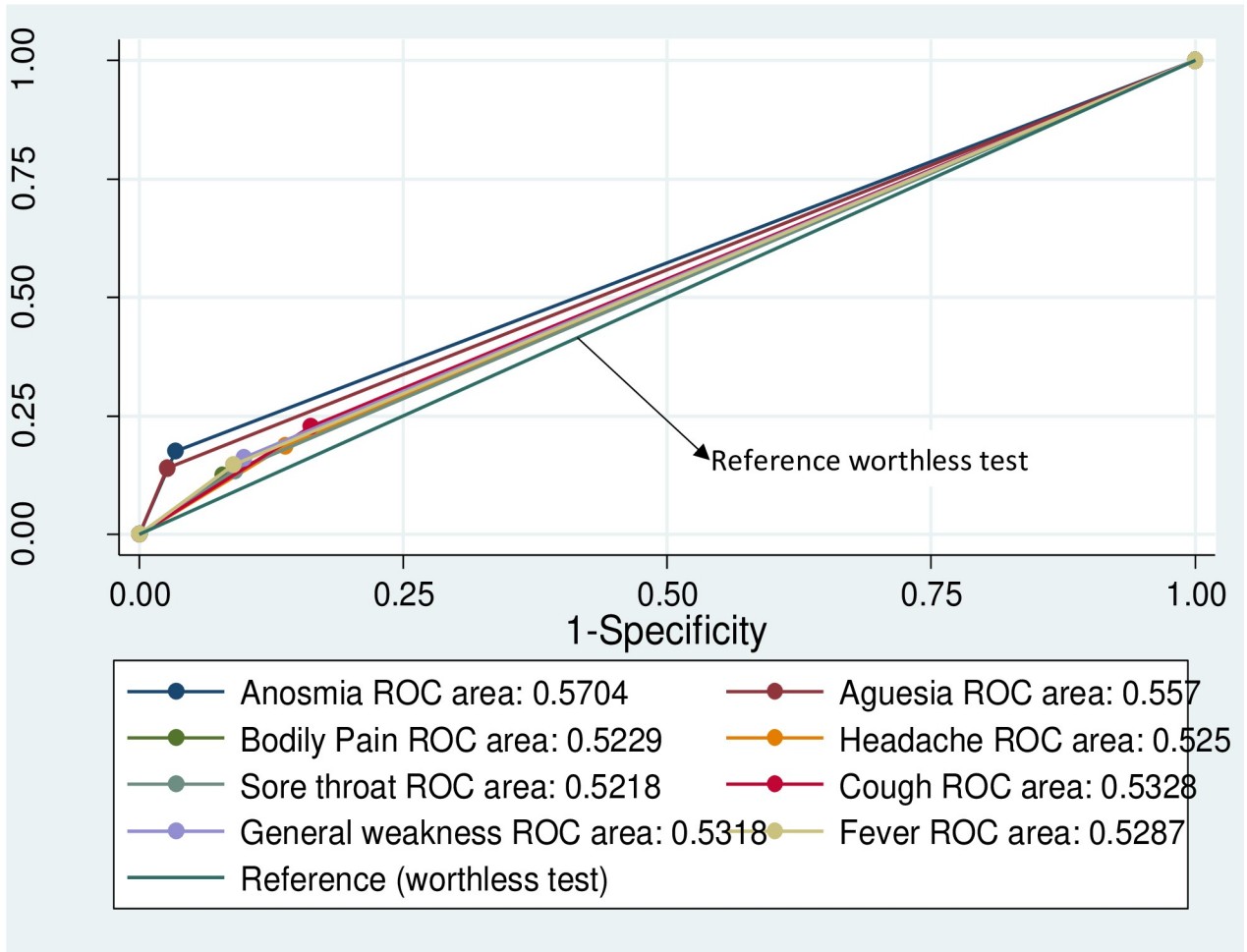

**Fig 3. Receiver operator characteristic curve for independent classifiers associated with SARS-CoV-2.**

Earlier studies have found symptoms such as loss of smell and taste as a strong predictor of COVID-19 [45]. Our study also found loss of smell and taste to be strongly associated with COVID-19, however, using loss of smell, loss of taste independently or in combinations failed to give a desirable test performance in our setting.

Symptoms questionnaire that ask for fever, cough, runny nose, headache, myalgia, diarrhoea among other symptoms do not perform differently from flipping of a coin in discriminating SARS-Cov-2 infected persons from uninfected persons. This could be attributed to the significantly high proportion of asymptomatic or pre-symptomatic infections as confirmed by this study where we found out that, 57.4% of all infected were asymptomatic at time of testing. This is consistent with other studies that found that, approximately half of those tested are asymptomatic or pre-symptomatic on the day of test [46].

Asymptomatic infections have several effects on efforts to contain the pandemic. Firstly, it renders symptom based screening that would be a good tool for rapid diagnosis and isolation to contain spread ineffective [47]. Also, persons with asymptomatic infections are able to go about their daily activities without assuming the sick role. The effect is that, asymptomatic persons spread the infections further [48, 49]. Due to the asymptomatic nature, most of such infections may go undiagnosed. Using a screening test associated with significant number of

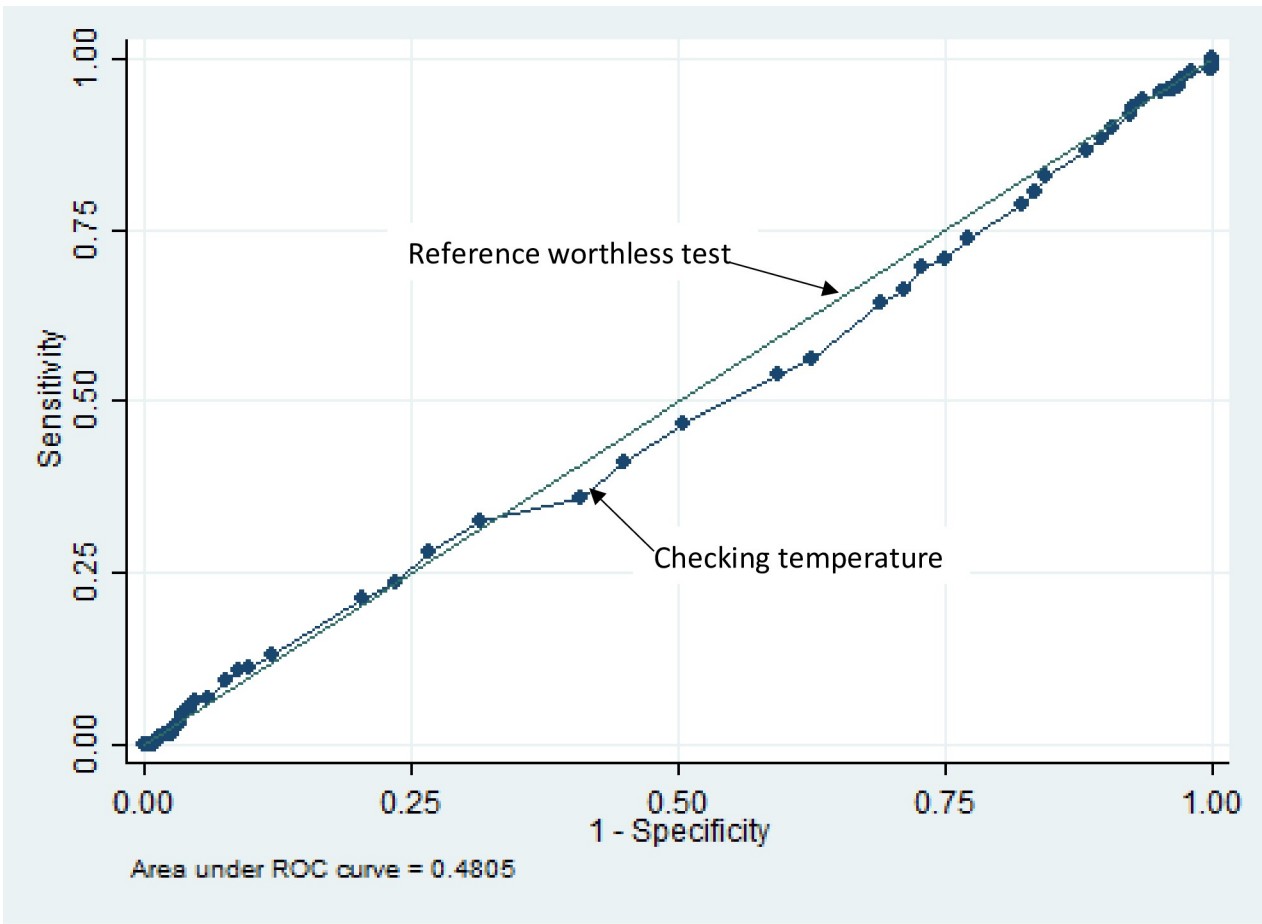

**Fig 4. Receiver operator characteristic curve for checked temperature as a screening tool for SARS-CoV-2 infection.**

false negatives, gives a false sense of safety to false negatives. This has the potential to promote unhealthy behaviour among persons misclassified as negatives because of the false belief that they are uninfected.

**Table 5. Receiver operator characteristics of combination of symptoms.**

| Case definition | | ALL Participants (symptomatic + Asymptomatic) | | | | | | Only Symptomatic (n = 663) | | | | | |
|---|---|---|---|---|---|---|---|---|---|---|---|---|---|
| | | Sensitivity | Specificity | Correctly classified | LR + | LR - | ROC Area [95% CI] | Sensitivity | Specificity | Correctly classified | LR + | LR - | ROC Area [95% CI] |
| | N | (%) | (%) | (%) | | | | (%) | (%) | (%) | | | |
| Tamale Teaching Hospital updated case definition (July 2020) | | | | | | | | | | | | | |
| Clinical criteria 1: Any two of; Fever, Cough, Sneezing, sore throat, Runny nose | 1963 | 17.5 | 88.1 | 76.2 | 1.5 | 0.94 | 0.53 [0.51–0.55] | 37.6 | 64.5 | 58.8 | 1.1 | 1.0 | 0.51 [0.47–0.56] |
| Clinical criteria 2: Any one of: difficulty in breathing, anosmia, ageusia | 1782 | 24.7 | 89.7 | 78.1 | 2.4 | 0.84 | 0.57 [0.55–0.60] | 46.7 | 73.4 | 67.3 | 1.8 | 0.7 | 0.60 [0.55–0.65] |
| Tamale Teaching Hospital Criteria 1 or 2 | 1781 | 33.8 | 81.9 | 73.3 | 1.9 | 0.81 | 0.58 [0.55–0.61] | 66.4 | 50.5 | 54.1 | 1.3 | 0.7 | 0.59 [0.53–0.63] |

This study had some limitations. It was not able to follow-up the asymptomatic persons or presumed pre-symptomatic persons to ascertain if they actually developed symptoms along the course of the ailment. This would be useful to determine the course of COVID-19 in sub-Saharan Africa. In the absence of a better screening alternative, the practice may continue amid caution to the public of its significant number of false negatives. Also, point of care rapid diagnostic tests need to be urgently deployed. There is the need for mass testing and appropriate isolation of confirmed positives so as to contain the outbreak. There is the need to scale up the number of facilities that can test so as to reduce the pressure on the few testing sites and improve the turnaround time for rT-PCR results.

## Conclusions

In conclusion, the use of temperature as a screening tool for SARS-CoV-2 infection is not worthwhile based on its test performance in ROC analysis. The use of fever alone or other symptoms individually [or in combination] as a screening tool for SARS-CoV-2 infection is not worthwhile based on ROC analysis. Shops and public areas that rely on non-contact thermometer temperature checking to grant access to public space need to redefine their strategies and rather insist on proven effective measures such as social distancing, wearing of face mask and hand hygiene practices.

## Supporting information

**S1 Table. ROC showing the performance of temperature screening for SARS-CoV-2 infection at given age range.**
(DOCX)

**S2 Table. Proportion of participants meeting the Tamale teaching hospital, Ghana; updated case definition (July 2020).**
(DOCX)

**S1 Fig. Receiver operator characteristic curve of clinical combination of symptoms of participants meeting the TTH updated case definition.**
(TIF)

**S1 Text. Tamale teaching hospital; updated COVID-19 case definition.**
(DOCX)

## Acknowledgments

We are grateful to the management of Tamale Teaching Hospital and the COVID-19 case management team for the support in caring for our patients. We are also grateful to Prof. Rajesh Vedanthan, Department of Population Health and Department of Medicine, NYU Langone Health, New York, New York, USA, for reviewing this manuscript and offering useful recommendations.

## Author Contributions

**Conceptualization:** Benjamin Demah Nuertey, Kwame Ekremet, Abdul-Rashid Haidallah, Kareem Mumuni, Joyce Addai, Rosemary Ivy E. Attibu, Michael C. Damah, Elvis Duorinaa, Anwar Sadat Seidu, Victor C. Adongo, Richard Kujo Adatsi, Hisyovi Caedenas Suri, Abass Abdul-Karim Komei, Braimah Baba Abubakari, Enoch Weyori, Emmanuel Allegye-Cudjoe, Richard O. Phillips.

**Data curation:** Benjamin Demah Nuertey, Kwame Ekremet, Abdul-Rashid Haidallah, Rosemary Ivy E. Attibu, Michael C. Damah, Elvis Duorinaa, Anwar Sadat Seidu, Victor C. Adongo, Richard Kujo Adatsi, Hisyovi Caedenas Suri, Abass Abdul-Karim Komei, Enoch Weyori, Emmanuel Allegye-Cudjoe, Augustina Sylverken, Michael Owusu, Richard O. Phillips.

**Formal analysis:** Benjamin Demah Nuertey, Joyce Addai.

**Funding acquisition:** Benjamin Demah Nuertey.

**Investigation:** Benjamin Demah Nuertey.

**Methodology:** Benjamin Demah Nuertey, Joyce Addai.

**Project administration:** Benjamin Demah Nuertey, Abdul-Rashid Haidallah.

**Resources:** Benjamin Demah Nuertey.

**Software:** Michael C. Damah.

**Supervision:** Benjamin Demah Nuertey, Kareem Mumuni, Joyce Addai, Elvis Duorinaa, Anwar Sadat Seidu, Hisyovi Caedenas Suri, Abass Abdul-Karim Komei, Braimah Baba Abubakari, Emmanuel Allegye-Cudjoe, Augustina Sylverken, Michael Owusu, Richard O. Phillips.

**Validation:** Richard Kujo Adatsi, Emmanuel Allegye-Cudjoe, Augustina Sylverken, Michael Owusu.

**Writing – original draft:** Benjamin Demah Nuertey, Joyce Addai.

**Writing – review & editing:** Benjamin Demah Nuertey, Kwame Ekremet, Abdul-Rashid Haidallah, Kareem Mumuni, Joyce Addai, Rosemary Ivy E. Attibu, Michael C. Damah, Elvis Duorinaa, Anwar Sadat Seidu, Victor C. Adongo, Richard Kujo Adatsi, Hisyovi Caedenas Suri, Abass Abdul-Karim Komei, Braimah Baba Abubakari, Enoch Weyori, Emmanuel Allegye-Cudjoe, Augustina Sylverken, Michael Owusu, Richard O. Phillips.

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
