## [Decision Letter · Decision Letter 0]

23 Jun 2021

PONE-D-21-14457

Performance of COVID-19 associated symptoms and temperature checking as a screening tool for SARS-CoV-2 infection

PLOS ONE

Dear Dr. Nuertey,

Thank you for submitting your manuscript to PLOS ONE. After careful consideration, we feel that it has merit but does not fully meet PLOS ONE’s publication criteria as it currently stands. Therefore, we invite you to submit a revised version of the manuscript that addresses the points raised during the review process.

The reviewer recommends that you make minor revisions to your work. Please ensure that previously published work in the same field are well acknowledged in the background of your manuscript and comparisons made. 

We look forward to receiving your revised manuscript.

Kind regards,

Martin Chtolongo Simuunza, PhD

Academic Editor

PLOS ONE

Journal Requirements:

3. We note that Figure 1 in your submission contain map images which may be copyrighted. All PLOS content is published under the Creative Commons Attribution License (CC BY 4.0), which means that the manuscript, images, and Supporting Information files will be freely available online, and any third party is permitted to access, download, copy, distribute, and use these materials in any way, even commercially, with proper attribution. For these reasons, we cannot publish previously copyrighted maps or satellite images created using proprietary data, such as Google software (Google Maps, Street View, and Earth). For more information, see our copyright guidelines: http://journals.plos.org/plosone/s/licenses-and-copyright.

3.1.    You may seek permission from the original copyright holder of Figure 1 to publish the content specifically under the CC BY 4.0 license. 

3.2.    If you are unable to obtain permission from the original copyright holder to publish these figures under the CC BY 4.0 license or if the copyright holder’s requirements are incompatible with the CC BY 4.0 license, please either i) remove the figure or ii) supply a replacement figure that complies with the CC BY 4.0 license. Please check copyright information on all replacement figures and update the figure caption with source information. If applicable, please specify in the figure caption text when a figure is similar but not identical to the original image and is therefore for illustrative purposes only.

5. Please provide additional details regarding participant consent. In the ethics statement in the Methods and online submission information, please ensure that you have specified (1) whether consent was informed and (2) what type you obtained (for instance, written or verbal, and if verbal, how it was documented and witnessed). If your study included minors, state whether you obtained consent from parents or guardians. If the need for consent was waived by the ethics committee, please include this information.

Reviewers' comments:

Reviewer's Responses to Questions

**Comments to the Author**

1. Is the manuscript technically sound, and do the data support the conclusions?

Reviewer #1: Yes

2. Has the statistical analysis been performed appropriately and rigorously? 

Reviewer #1: Yes

3. Have the authors made all data underlying the findings in their manuscript fully available?

Reviewer #1: Yes

4. Is the manuscript presented in an intelligible fashion and written in standard English?

Reviewer #1: Yes

5. Review Comments to the Author

Reviewer #1: The manuscript submitted by Dr. B. Demah Nuertey et al. aims to determine the usefulness of clinical symptoms in accurately predicting a final diagnosis of COVID-19. The manuscript is clear, well organized, and well written. Authors used a dataset of ~2,000 participants, from Tamale teaching hospital, Ghana. The study is generally sound, authors assess associations between symptoms and PCR positive test result using a multivariate logistic framework, considering the possible correlation between symptoms. Authors further performed a ROC analysis.

Authors' conclusion is that "The use of fever alone or other symptoms individually as a screening tool for SARS-CoV-2 infection is not worthwhile based on ROC analysis."

Major comments:

- While the usefulness of temperature and other symptoms in COVID-19 screening is a key question, it is important acknowledge that other studies have been conducted, and replace the study in the context of the current literature (e.g. among others, a study in Australia https://onlinelibrary.wiley.com/doi/10.1111/1742-6723.13578 for temperature check, and a meta-analysis https://gut.bmj.com/content/70/4/806#ref-1 for loss of smell/taste).

- A more in-depth exploration of the dataset could strengthen the manuscript. For example: is the temperature screening performing differently given the age of the patient? Would other symptoms be interesting to consider in the screening? Would a screening of a combination of symptoms perform better than individual ones? Also, it would be further interesting to place this study in a global context and see if there are any key differences in terms of disease presentation and/or prevalence of different symptoms.

Minor comments:

- In the abstract, the sentence: ‘The likelihood of a positive test result for SARS-CoV-2 was 5.9 [3.9 – 8.8] for loss of sense of smell and 5.9 [3.8 – 9.3] for loss of sense of taste.’ is unclear. A likelihood should be a probability, here it seems to be a relative risk of presenting with the symptoms when being positive, compared to when being negative. It would be better to use the formulation from page 10, which is well written.

- In the methods, page 6, it would be useful to see the CT threshold used to declare the RT-PCR as positive.

- In the methods, page 6, the term ‘Real-time RT-PCR’ is redundant.

- The reporting of the ROC analysis from Table 4 is inaccurate: 4.8% sensitivity corresponds to a temperature cut-point of 37.7, and not 37.8.

6. PLOS authors have the option to publish the peer review history of their article (what does this mean?). If published, this will include your full peer review and any attached files.

Reviewer #1: No

---

## [Author Response · Author response to Decision Letter 0]

2 Jul 2021

30th June, 2021

The editor

PLOS one

Dear Editor,

Response to Reviewers: Performance of COVID-19 associated symptoms and temperature checking as a screening tool for SARS-CoV-2 infection

Thank you for the opportunity and improving the quality of this manuscript. Below is how we addressed the comments of the reviewers.

Journal/Editorial requirement

• The data set has been made public and can be found here https://osf.io/fz82k/

• Also the map was changed using suggested resources

Reviewer #1: 

comment

- While the usefulness of temperature and other symptoms in COVID-19 screening is a key question, it is important acknowledge that other studies have been conducted, and replace the study in the context of the current literature (e.g. among others, a study in Australia https://onlinelibrary.wiley.com/doi/10.1111/1742-6723.13578 for temperature check, and a meta-analysis https://gut.bmj.com/content/70/4/806#ref-1 for loss of smell/taste).

Response

Thank you for the suggestions which significantly improved the manuscript. The abstract and introduction were updated with findings from current literature including the suggested literature to reflect the context

comments

- A more in-depth exploration of the dataset could strengthen the manuscript. For example: is the temperature screening performing differently given the age of the patient? Would other symptoms be interesting to consider in the screening? Would a screening of a combination of symptoms perform better than individual ones? Also, it would be further interesting to place this study in a global context and see if there are any key differences in terms of disease presentation and/or prevalence of different symptoms.

Response

Thank you very much, suggestions were carried out and a more in-depth exploration of the dataset was carried out. The performance of the temperature check at varying age groups were carried out and shown in supplement table 1. Also, a combination of symptoms such as the case definition used for screening in the Tamale teaching Hospital, Ghana attached as supplement table 2 was explored. A section on the combination of symptoms was added to the results and the performance of the combination of symptoms added as table 5.

Minor comments:

comment

- In the abstract, the sentence: ‘The likelihood of a positive test result for SARS-CoV-2 was 5.9 [3.9 – 8.8] for loss of sense of smell and 5.9 [3.8 – 9.3] for loss of sense of taste.’ is unclear. A likelihood should be a probability, here it seems to be a relative risk of presenting with the symptoms when being positive, compared to when being negative. It would be better to use the formulation from page 10, which is well written.

Response: Thank you very much, your suggestion was accepted and the formulation in page 10 was used as shown in the abstract, results section lines 56-57

Comment

- In the methods, page 6, it would be useful to see the CT threshold used to declare the RT-PCR as positive.

Response:

The following was included in the method section of the manuscript to address your suggestion. “Samples with SARS-CoV-2 RT PCR cycle threshold (CT) value under 40 were considered positive”

Comment

- In the methods, page 6, the term ‘Real-time RT-PCR’ is redundant.

Response:

Thank you, “Real-time” was deleted

Comment

- The reporting of the ROC analysis from Table 4 is inaccurate: 4.8% sensitivity corresponds to a temperature cut-point of 37.7, and not 37.8.

Response

Thank you, the error was corrected; 4.8% was changed to 4.2%. the sentence now reads; “For example, using a temperature cut off of 37.8 would only pick 4.2% of positive SARS-CoV-2 infected patients”

Once again thank you for the suggestions which greatly improved the manuscript

Yours faithfully,

Dr. Benjamin Nuertey

---

## [Decision Letter · Decision Letter 1]

26 Jul 2021

PONE-D-21-14457R1

Performance of COVID-19 associated symptoms and temperature checking as a screening tool for SARS-CoV-2 infection

PLOS ONE

Dear Dr. Nuertey,

Thank you for submitting your manuscript to PLOS ONE. After careful consideration, we feel that it has merit but does not fully meet PLOS ONE’s publication criteria as it currently stands. Therefore, we invite you to submit a revised version of the manuscript that addresses the points raised during the review process.

The reviewer recommends that you include some discussion for the analysis that you added in the manuscript. Please attend to the concerns that have been raised and then return the revised manuscript as advised in this letter.

We look forward to receiving your revised manuscript.

Kind regards,

Martin Chtolongo Simuunza, PhD

Academic Editor

PLOS ONE

Journal Requirements:

Reviewers' comments:

Reviewer's Responses to Questions

**Comments to the Author**

1. If the authors have adequately addressed your comments raised in a previous round of review and you feel that this manuscript is now acceptable for publication, you may indicate that here to bypass the “Comments to the Author” section, enter your conflict of interest statement in the “Confidential to Editor” section, and submit your "Accept" recommendation.

Reviewer #1: (No Response)

2. Is the manuscript technically sound, and do the data support the conclusions?

Reviewer #1: Yes

3. Has the statistical analysis been performed appropriately and rigorously? 

Reviewer #1: Yes

4. Have the authors made all data underlying the findings in their manuscript fully available?

Reviewer #1: Yes

5. Is the manuscript presented in an intelligible fashion and written in standard English?

Reviewer #1: Yes

6. Review Comments to the Author

Reviewer #1: The paper has been greatly improved with more information on the context of the study and additional analyses, I thank the authors for their response. However, some minor issues remain, mainly on the discussion. I will mention some further below for the authors to consider.

Comments:

- The analysis of the performance on the temperature screening by age is a nice add-on, but it is not discussed at all. Also, the ‘desirable’ performance is not defined in the main text, and in Supplementary Table 1 the number of patients considered in each age group is not given.

- The analysis of the performance of the combination of symptoms is very interesting. However, I would consider rephrasing the paragraph ‘Combination of symptoms’, as it is not clearly written. Some key figures would improve the reading. Further, these results are not discussed either. Discussing these results, side by side with the temperature check would improve the paper and strengthen the authors’ result that “The use of fever alone or other symptoms individually [or in combination] as a screening tool for SARS-CoV-2 infection is not worthwhile based on ROC analysis”. Also, it could be noted that the case definition of the Tamale hospital (criteria 1 or 2) led to quite a lot of missed cases.

7. PLOS authors have the option to publish the peer review history of their article (what does this mean?). If published, this will include your full peer review and any attached files.

Reviewer #1: No

---

## [Author Response · Author response to Decision Letter 1]

10 Aug 2021

RE: PONE-D-21-14457R1: Performance of COVID-19 associated symptoms and temperature checking as a screening tool for SARS-CoV-2 infection. 

Thank you for the opportunity to provide a revision to this work. 

Response to reviewer

We are grateful to the editor and the reviewer for your determination to improve this manuscript. This kind gesture is well appreciated. We are privileged to have you review this manuscript. Below is how we address your comment;

Comments:

The paper has been greatly improved with more information on the context of the study and additional analyses, I thank the authors for their response. However, some minor issues remain, mainly on the discussion. I will mention some further below for the authors to consider.

- The analysis of the performance on the temperature screening by age is a nice add-on, but it is not discussed at all. Also, the ‘desirable’ performance is not defined in the main text, and in Supplementary Table 1 the number of patients considered in each age group is not given.

Response

Discussion on temperature screening by age has been included in the discussion now. We are grateful for pointing this out to us.

Desirable performance defined: the following was added to the method, statistical analysis, page 8, lines 200-202 “ The performance of a screening test is deemed to be desirable if Area Under Curve (AUC) is 0.7 � AUC � 0.8, excellent if 0.8 < AUC � 0.9 and outstanding if AUC is > 90 [37].”

In Supplementary table 1, the number of patients considered and corresponding percentage in each age group has been provided now.

Comment

- The analysis of the performance of the combination of symptoms is very interesting. However, I would consider rephrasing the paragraph ‘Combination of symptoms’, as it is not clearly written. Some key figures would improve the reading. Further, these results are not discussed either. Discussing these results, side by side with the temperature check would improve the paper and strengthen the authors’ result that “The use of fever alone or other symptoms individually [or in combination] as a screening tool for SARS-CoV-2 infection is not worthwhile based on ROC analysis”. Also, it could be noted that the case definition of the Tamale hospital (criteria 1 or 2) led to quite a lot of missed cases

Response

Paragraph on combination of symptoms was rephrased and re-written. Some key figures were included to improve the reading. Page 14 to 15, lines 268 - 293 shows this modification. 

Some key figures to improve reading: supplemental figures 1 and 2 had been moved to main manuscript and named Figures 3 and 4 respectively. Also, a new figure displaying the three ROC curves for clinical criteria 1, 2 and combination of 1 or 2 had been included as supplemental figure 1. 

Also, it was noted in the results section, page 15, lines 291 – 293 that, the case definition of the Tamale hospital (criteria 1 or 2) led to quite a lot of missed cases. “The combination of symptoms as exemplified by clinical criteria 1 or 2 missed 66.3% of all positive cases.”

Discussion section has been updated to include reviewer suggestions.

As part of the update on the discussions and to address reviewer comment, the following publications were cited and referenced appropriately.

• Mandrekar JN. Receiver operating characteristic curve in diagnostic test assessment. Journal of Thoracic Oncology. 2010;5:1315–6.

• Bielecki M, Crameri GAG, Schlagenhauf P, Buehrer TW, Deuel JW. Body temperature screening to identify SARS-CoV-2 infected young adult travellers is ineffective. Travel Med Infect Dis. 2020;37:101832. doi:10.1016/j.tmaid.2020.101832.

• Wright WF, Mackowiak PA. Why Temperature Screening for Coronavirus Disease 2019 With Noncontact Infrared Thermometers Does Not Work. Open Forum Infectious Diseases. 2021;8. doi:10.1093/ofid/ofaa603.

• Roland LT, Gurrola JG, Loftus PA, Cheung SW, Chang JL. Smell and taste symptom-based predictive model for COVID-19 diagnosis. International Forum of Allergy & Rhinology. 2020;10:832–8. doi:10.1002/alr.22602.

---

## [Decision Letter · Decision Letter 2]

2 Sep 2021

Performance of COVID-19 associated symptoms and temperature checking as a screening tool for SARS-CoV-2 infection

PONE-D-21-14457R2

Dear Dr. Nuertey,

We’re pleased to inform you that your manuscript has been judged scientifically suitable for publication and will be formally accepted for publication once it meets all outstanding technical requirements.

Kind regards,

Martin Chtolongo Simuunza, PhD

Academic Editor

PLOS ONE

Additional Editor Comments (optional):

Reviewers' comments:

Reviewer's Responses to Questions

**Comments to the Author**

1. If the authors have adequately addressed your comments raised in a previous round of review and you feel that this manuscript is now acceptable for publication, you may indicate that here to bypass the “Comments to the Author” section, enter your conflict of interest statement in the “Confidential to Editor” section, and submit your "Accept" recommendation.

Reviewer #1: All comments have been addressed

2. Is the manuscript technically sound, and do the data support the conclusions?

Reviewer #1: Yes

3. Has the statistical analysis been performed appropriately and rigorously? 

Reviewer #1: Yes

4. Have the authors made all data underlying the findings in their manuscript fully available?

Reviewer #1: Yes

5. Is the manuscript presented in an intelligible fashion and written in standard English?

Reviewer #1: Yes

6. Review Comments to the Author

Reviewer #1: The authors have suitably addressed my comments. I believe the changes made a substantial improvement to the paper, so I recommend publication.

7. PLOS authors have the option to publish the peer review history of their article (what does this mean?). If published, this will include your full peer review and any attached files.

Reviewer #1: No

---

## [Editor Report · Acceptance letter]

8 Sep 2021

PONE-D-21-14457R2 

Performance of COVID-19 associated symptoms and temperature checking as a screening tool for SARS-CoV-2 infection 

Dear Dr. Nuertey:

I'm pleased to inform you that your manuscript has been deemed suitable for publication in PLOS ONE. Congratulations! Your manuscript is now with our production department. 

Kind regards, 

on behalf of

Dr. Martin Chtolongo Simuunza 

Academic Editor

PLOS ONE